# Students’ Well-Being and Academic Engagement: A Multivariate Analysis of the Influencing Factors

**DOI:** 10.3390/healthcare12151492

**Published:** 2024-07-27

**Authors:** Silvia Puiu, Mihaela Tinca Udriștioiu, Iulian Petrișor, Sıdıka Ece Yılmaz, Miriam Spodniaková Pfefferová, Zhelyazka Raykova, Hasan Yildizhan, Elisaveta Marekova

**Affiliations:** 1Department of Management, Marketing and Business Administration, Faculty of Economics and Business Administration, University of Craiova, 200585 Craiova, Romania; 2Department of Physics, Faculty of Sciences, University of Craiova, 200585 Craiova, Romania; mtudristioiu@central.ucv.ro (M.T.U.); iulian.petrisor@edu.ucv.ro (I.P.); 3Career Planning Application and Research Center, Adana Alparslan Türkeş Science and Technology University, 46278 Adana, Türkiye; eyilmaz@atu.edu.tr; 4Department of Physics, Faculty of Natural Sciences, Matej Bel University, 97401 Banská Bystrica, Slovakia; miriam.spodniakova@umb.sk; 5Department of Educational Technologies, Faculty of Physics and Technology, University of Plovdiv Paisii Hilendarski, 4000 Plovdiv, Bulgaria; janeraik@uni-plovdiv.bg (Z.R.); eligeo@uni-plovdiv.bg (E.M.); 6Energy Systems Engineering, Engineering Faculty, Adana Alparslan Türkeş Science and Technology University, 46278 Adana, Türkiye; hyildizhan@atu.edu.tr

**Keywords:** well-being, social support, satisfaction, academic engagement

## Abstract

This paper aims to identify the factors that are positively or negatively impacting students’ well-being and their academic engagement. We used partial least-squares structural equation modeling (PLS-SEM) using the data collected through a questionnaire from four countries: Romania, Turkey, Slovakia, and Bulgaria. The model includes seven factors that influence the well-being of students and indirectly their academic engagement: stressors in the students’ lives; professors’ support; social support from family and friends; the students’ perceived satisfaction in their lives; engaging in activities during their leisure time; self-exploration regarding their careers; and environmental exploration regarding their careers. The results show that all factors, except for stressors and environmental exploration regarding their careers, positively influence the students’ well-being and thus their academic engagement. These findings are useful for university professors and managers in better organizing activities to increase academic performance.

## 1. Introduction

The present paper investigates the role played by several factors in students’ well-being and academic engagement in four countries: Romania, Turkey, Slovakia, and Bulgaria. The topic of well-being should be carefully considered and integrated into educational strategies considering its role in academic results [1]. Turner et al. [2] (p. 707) appreciate that “universities can actively support student well-being by fostering resilience”, and this can be carried out in several ways such as changing the structure of the courses or training the professors and the administrative staff to communicate efficiently with students. Other authors [3] emphasize the importance of professors in the well-being of students, considering several factors such as their communication skills, their attitudes, and the support they offer to their students. Baik et al. [4] appreciate that mental health as a part of well-being should be “a critical issue” for universities which have to understand the need to strengthen the partnership between students and the other actors in the system (professors, management, and administrative staff).

Jones et al. [5] also reflect on the importance of the well-being of students as a key element that should be incorporated into the universities’ strategies. The authors of [5] (p. 438) appreciate that well-being should be “part of a wider whole university approach” and included in the way professors structure their courses and lectures, evaluate assessments, and in general in how they communicate and offer support. Egan et al. [6] (p. 301) highlighted the relationship between academic performance and well-being, resiliency, and self-compassion, mentioning the need for “appropriate interventions which are user friendly, affordable and can be embedded into existing student learning and support”. 

Our research focuses on a multitude of factors that can influence students’ well-being and their academic results. These factors are the stressors in the students’ lives (related to family problems, financial burden, or stressors related directly to the university environment); the support received from their professors, family, and friends; the satisfaction felt by students with the university; the activities they engage in during their leisure time (outside school); and career exploration (self-exploration and environmental exploration). 

A novel aspect of our research is the fact that we combined a multitude of factors affecting well-being and academic engagement to see which are the ones with the highest impact. Hook and Bogdanov [7] state in their study the way mental health issues (which are part of well-being) encounter a negative perception in Eastern European countries. The same stigma is highlighted by other authors [8] for Turkey. This research motivated us to conduct a study in the four countries to raise the level of awareness regarding the importance of well-being (with all its dimensions) in the academic performance desired by all universities (starting from management in higher education institutions to the professors engaging with students). 

## 2. Literature Review

### 2.1. Students’ Well-Being

The interest in students’ well-being has increased in recent years, as well as in the context of the COVID-19 pandemic which posed many challenges to the educational process and all educational actors (students, professors, administrative staff, and management in educational institutions). Barbayannis et al. [9] emphasize the need to offer support for students who experience a decline in their well-being due to the stress felt in crises such as the pandemic. Klapp et al. [10] conclude that there is a positive relationship between well-being and the academic achievement of students and that the “causes of the decrease in well-being may be changes in the educational and assessment system”, which shows the role played by professors in the way they evaluate students or the strategies implemented by the academic management. The same correlation between well-being and academic performance was noticed by Joseph et al. [11] for students in India. A similar study was conducted for students in China; Zhang et al. [12] noticed that the use of social media affects the students’ well-being and also that a high level of well-being improves their academic results. Supranowicz and Paz [13] refer to the three dimensions of well-being: physical well-being, mental or psychological well-being, and social well-being. 

### 2.2. The Stressors in Students’ Lives

There are many factors of stress for students, ranging from a normal level, inherent before exams or other assessments, to the stress that might negatively affect the well-being and the academic engagement of students, such as financial problems, a high volume of assignments from the professors, and problems related to family or not having social support. Usually, when referring to stress, most papers study the stress that exceeds normal limits and that can have a strong and negative impact on someone’s life. Thus, Slimmen et al. [14] emphasize the impact stress has on mental well-being which is important because several stressors might have a different impact and also a different approach for increasing students’ resilience and their coping skills. 

Ocaña-Moral et al. [15] studied the effect of stress on life satisfaction and academic performance for students in Spain in a year in which the pandemic was still a problem. Their results show that the impact is stronger concerning satisfaction but the stress did not have a significant impact on students’ performance. Bono et al. [16] focused on the approaches that can be used to reduce the impact of stress on the well-being of students and showed that gratitude can be a great instrument. This is useful for academic management and professors who might use this and other tools to help students be more resilient and cope better in difficult times or times that are more stressful than usual. Denovan and Macaskill [17] highlight the importance of offering students the “interventions to develop optimism”, seen as a solution for coping with stress. Carvalho et al. [18] emphasize the relationship between positive youth development, stress, and well-being and the differences that arise in adolescence between girls and boys which might explain their resilience capacity when faced with stressful events. 

### 2.3. The Professors’ Support 

Brandseth et al. [19] show that the support offered by professors to their students is positively linked to both the feeling of belonging to the class and their well-being. The authors emphasize the importance of including these findings in educational strategies and “promoting teachers’ supportive behavior”. Suldo et al. [20] state that students appreciate more the emotional support offered by their professors and their efforts to create a climate in which they feel encouraged to address questions and have a debate on the topics discussed. Riva et al. [21] emphasize the importance of the relationship between professors and students in higher education institutions for both the well-being of professors and their students. Soini et al. [22] studied the role played by the social interaction of the professors with their students for the professors’ well-being. So, offering adequate support (including emotional support) for helping students in challenging situations helps professors increase their well-being. 

### 2.4. Social Support 

In the study conducted by Holliman et al. [23] (p. 1) on students and non-students in the United Kingdom, “social support was found to make a significant independent contribution to most wellbeing outcomes”. Zeidner et al. [24] showed that social support had a greater impact on well-being than coping, with the authors appreciating it as being “critical”. Zimet et al. [25] appreciate that social support comes mostly from three sources: family, friends, and significant others. Poots and Cassidy [26] found that social support increases well-being and decreases the stress felt by students. Malkoç and Yalçin [27] (p. 35) researched Turkish students and noticed the same correlation between support from family and friends and the students’ well-being, with the social support also mediating “the relationship between resilience and psychological well-being”. 

### 2.5. The Students’ Perceived Satisfaction in Their Lives 

Studies on the satisfaction felt by students reveal the impact it has on their well-being and academic performance. As Vermunt et al. [28] (p. 1) state, satisfaction perceived by an individual is reflected by “the discrepancy between the situation one has and the situation one aspires to”. Urquijo et al. [29] found a direct and positive impact of students’ emotional intelligence and satisfaction on their levels of well-being and a negative correlation between stress and their perceived satisfaction. Ruiz-Aranda et al. [30] researched students in the health area and noticed that their satisfaction is influenced by their emotional intelligence, contributing in this way to their well-being. The authors highlight the need to include in the curricula subjects that teach students better ways of coping and that can enhance their emotional intelligence and, in the end, lead to higher levels of satisfaction and well-being. Even if this was a study addressed to future professionals in the health sector, the results and recommendations can be applied to other universities too, considering the role played by well-being in the academic performance of students. 

### 2.6. Engaging in Activities during Their Leisure Time 

Congsheng et al. [31] studied students in Malaysia and found that their mental well-being is positively influenced by the sports activities and other physical activities in which they are engaged, which is useful for academic management in universities that can create more opportunities for students, both curricular and extra-curricular. Trupp [32] highlighted that volunteering helps students have a higher well-being, and this was noticed in long-term engagement in volunteering activities. A similar correlation between physical activity and well-being was noticed by Molina-Garcia et al. [33] for Spanish students with some slight differences between men and women. The authors emphasized that higher levels of physical activity led to a higher level of well-being. 

Jaskulska et al. [34] studied the relationship between leisure time activities and well-being in Polish adolescents during the COVID-19 pandemic and noticed a positive correlation mostly for boys who were engaging in sports activities more than girls who preferred to socialize. Doerksen et al. [35] notice that the type of activity and time allocated to that activity are also important in influencing students’ well-being. The activities that generated higher levels of well-being were social and volunteering activities. Trainor et al. [36] noticed a difference between structured and unstructured leisure time activities, with those more structured having a higher impact on students’ well-being compared with the others. 

### 2.7. Self-Exploration and Environmental Exploration Regarding Their Careers 

Both of these dimensions are part of the career exploration survey proposed by Stumpf et al. [37]. The difference between them is that for self-exploration, the students are engaged in a self-reflection process regarding their careers, while environmental exploration is oriented towards the environment and the career opportunities it provides for them. Lazarides et al. [38] found that career exploration (with both dimensions) is positively influenced by the students’ intrinsic motivation. Ketterson and Blustein [39] analyzed the relationship between the attachment style of students towards their parents and career exploration and noticed a positive correlation when the attachment was a secure one. Ran et al. [40] found a positive correlation between career exploration (with both dimensions) and the students’ well-being. Zhou et al. [41] highlight that career exploration is strongly influenced by the proactive personality of students and indirectly influences their well-being. 

### 2.8. Academic Engagement

Academic engagement is different from academic performance, revealing the efforts and the willingness of students to engage in various academic activities (curricular or extracurricular). Academic engagement is a process that can lead with time to higher academic performance. Datu and King [42] conclude that there is a reciprocal relationship between academic engagement and students’ well-being, with one leading to another and vice versa. Boulton et al. [43] researched UK students and found that academic engagement positively impacts their well-being, but the correlation between engagement and their academic results was a negative one, which can be partially explained by the decrease in the engagement levels towards the end of the semester (before the exams and other evaluations). A study on Iranian students revealed the same positive correlation between academic engagement and well-being [44]. Kotera et al. [45] researched students in the UK and Indonesia, finding a positive correlation between vigor (as a dimension of academic engagement) and their well-being, which was also influenced by self-compassion. Eriksen and Bru [46] state the strong link between emotional well-being and academic engagement, highlighting the importance of emotions in the educational process.

### 2.9. Development of Hypotheses 

Based on the literature review, we developed the following hypotheses:

**Hypothesis 1 (H1).** *The stressors in the students’ lives have a direct and negative impact on their well-being*.

**Hypothesis 2 (H2).** *The professors’ support has a direct and positive impact on students’ well-being*.

**Hypothesis 3 (H3).** *Social support has a direct and positive impact on students’ well-being*.

**Hypothesis 4 (H4).** *The students’ perceived satisfaction in their lives has a direct and positive impact on their well-being*.

**Hypothesis 5 (H5).** *Engaging in activities during their leisure time has a direct and positive impact on students’ well-being*.

**Hypothesis 6 (H6).** *Self-exploration regarding their careers has a direct and positive impact on students’ well-being*.

**Hypothesis 7 (H7).** *Environmental exploration regarding their careers has a direct and positive impact on students’ well-being*.

**Hypothesis 8 (H8).** *The students’ well-being has a direct and positive impact on their academic engagement*.

## 3. Materials and Methods

### 3.1. Participants

We conducted our study in Romania, Turkey, Slovakia, and Bulgaria. The decision to use these countries was based on the support we received within the Advtech_AirPollution project (Applying some advanced technologies in teaching and research about air pollution, 2021-1-RO01-KA220-HED-000030286) funded by the European Union within the framework of the Erasmus+ Program in which the universities from these countries were partners. 

The characteristics of the sample in terms of age, sex, level of studies, and employment status for each country are presented in Table 1. Most respondents were female students (61.3%) between 18 and 22 years old (60.4%) with a bachelor’s degree (75.6%) and unemployed (55.9%). There were slight differences between countries: for example, most students in Romania and Bulgaria had a job, compared with the students in Slovakia (61%) and Turkey (80.6%) who were mostly unemployed. 

### 3.2. Procedures and Measures

The questionnaire was built in Google Forms and sent to 2400 students (600 per country) during October 2023 and January 2024. We received answers from 1051 students: 418 students in Slovakia, 296 in Romania, 162 from Bulgaria, and 175 students from Turkey. The statements in the questionnaire were on a Likert scale from 1 (total disagreement) to 5 (total agreement). We used the non-probabilistic snowball sampling method. Each respondent gave his/her informed consent and no personal data were collected. The constructs and the items used in our research were included in the survey as statements, as illustrated in Table 2. 

### 3.3. Data Analysis

As a research methodology, we used partial least-squares structural equation modeling (PLS-SEM) and the software SmartPLS, version 4.1.0.0 [47]. The main objective of this research was to analyze the factors that influence the students’ well-being and their academic engagement in Romania, Turkey, Slovakia, and Bulgaria. We chose PLS-SEM because it is often preferred when the number of variables is higher and the complex model has many relationships [48]. Another reason is the small sample size, especially in Bulgaria and Turkey, where we had less than 200 respondents.

### 3.4. Theoretical Model

The research model developed in SmartPLS [47] is presented in Figure 1. The model comprised the variables presented in the literature review to have an impact on the students’ well-being, such as the support received from professors (PSPRT), social support (SSPRT), stressors (STRS), leisure time activities (LTA), their satisfaction (STSF), self-exploration and environmental exploration regarding their careers (CSE and CEE), and academic engagement as a dependent variable. The items in the model are detailed in Table 2. The model is novel because it includes multiple factors that can impact well-being and indirectly academic engagement. Another element of novelty is related to the inclusion of self-exploration and environmental exploration of the career choices in this complex model. It is helpful for academic managers, professors, and career counselors in universities. 

## 4. Results

In Table 3, we determined the outer loadings and the variance inflation factor (VIF) for the items in the model to check their reliability and collinearity.

All items with outer loadings above 0.7 were kept in the model, with the level indicating a high reliability. After carefully analyzing all items, we decided to keep those between 0.6 and 0.7 too, considering their importance and the fact that this range is still acceptable for the reliability of a model [49,50]. The new model is presented in Figure 2. After recalculating the outer loadings for the new model, the value of AENG3 decreased below the threshold and was finally eliminated. All VIF values are below 4, which ensures the collinearity of the proposed model. The strongest impact is from WB to AENG (0.561), followed by the impact from STSF to WB (0.34), and the least impact is from CEE to WB (0.016), from CSE to WB (0.081), and from STRS to WB (−0.036). WB generates 31.4% of the AENG variance, and STRS, PSPRT, SSPRT, CEE, CSE, STSF, and LTA generate 54.7% of the WB variance.

In Table 4, we determined the mean and the standard deviation for the items kept in the model that have outer loadings above 0.6. The highest means were registered for PSPRT4, with SSPRT 1–3 (above 4) showing the importance of the support received by students from their professors, family, and friends. The lowest means (below 2.5) were registered by some of the stressors (STRS3 related to the financial situation and STRS9 related to the conditions provided by the university).

The reliability and validity of the constructs in the model are determined and shown in Table 5. Cronbach’s alpha is higher than 0.7 for 8 of the 9 constructs, but the value for LTA is considered acceptable [51,52], being higher than 0.6 and close to 0.7. The same is noticed for the composite reliability. The recommendation is for these values to be higher than 0.7, a value reached for most constructs, except for LTA which registers a value below, but close to 0.7, with the level being considered acceptable to ensure the internal consistency of the model. The average variance extracted (AVE) is above 0.5 for all constructs, being an indicator of the validity of the model. The Standardized Root-Mean-Square Residual (SRMR) is 0.06, which is lower than the threshold of 0.08 for the model to be considered a good fit.

To check the discriminant validity of the constructs, we used the Fornell–Larcker criterion in Table 6. The values in the main diagonal are higher than the ones in the same column, which shows that the constructs in the model are sufficiently different from the others to ensure the discriminant validity.

Further, we applied the bootstrapping test in Table 7 to check for statistical significance. For the relation between STRS and WB and, respectively, between CEE and WB, the t values are below 1.96 and the p values are higher than 0.05. Thus, hypotheses H1 and H7 are not supported, which is also shown by the bias-corrected confidence intervals which includes the zero value.

## 5. Discussion

This research centers on a wide range and variety of factors that can impact students’ well-being and their academic outcomes. These factors are stressors in students’ lives; students’ well-being; professors’ support; social support from family, friends, and colleagues; the students’ perceived satisfaction in their lives; engaging in activities during their leisure time; self-exploration regarding their careers; environmental exploration regarding their careers; and the academic engagement of students. The results of the hypothesis are presented below.

The H1 hypothesis was not supported. The stressors in the students’ lives had a mean below 3, and the respondents disagreed with the statements related to the stressors in their lives. The only stressor with a value higher than 3 (3.49) was STRS6 (related to the many assignments given to them by professors). This might be attributed to the age of respondents (with 60.4% of them being between 18 and 22 years old) and the support they receive from their families at this age, a fact reflected by the validation of the H2 and H3 hypotheses. Most research papers show that youngsters have higher levels of stress that affect their well-being and performance [53,54,55,56]. Pascoe et al. [53] highlight the negative impact of stress on mental and physical health and Khan et al. [55] emphasize its influence on performance. Hook and Bogdanov [7] mention the stigma related to mental health problems in countries from Eastern Europe, which might partially explain why students in these countries might be reluctant to disclose their stress levels.

The H2 hypothesis was supported, and the professors’ support had a direct and positive impact on students’ well-being. This finding is important, showing the important role played by the attitude of professors towards their students beyond the formal education they offer. Professors can also be a stressor for students when they overwhelm them with too many assignments during the semester. In our study, this was the most important factor of stress for the respondents. This is consistent with the findings in other studies [20,57]. Suldo et al. [20] (p. 67) emphasize that “students perceive teachers to be supportive primarily when they attempt to connect with students on an emotional level”, encouraging them and offering them constructive feedback. Woloshyn et al. [57] (p. 82) state that students in Canada and Croatia reported higher levels of well-being if they “felt supported professionally and personally”.

The H3 hypothesis was supported. Thus, social support has a direct and positive impact on students’ well-being. The support received by students from their families, friends, and colleagues is important, influencing their well-being and indirectly their academic results. This finding is important because universities with the help of professors and other advisors should know if the student has difficulties in managing his or her activities due to a lack of proper support. Students in need might be offered adequate support after carefully examining the specifics of each situation. Other researchers reached similar conclusions [26,58,59,60]. Poots and Cassidy [26] emphasize the role played by social support in mediating “the relationship between academic stress and wellbeing”. Awang et al. [58] found that social support from family and friends helps students with their emotional adjustment, which is a reflection of their well-being. Alsubaie et al. [60] state the importance of social support for the quality of life, which is a predictor of their well-being.

The H4 hypothesis according to which the students’ perceived satisfaction in their lives has a direct and positive impact on their well-being was supported. Other authors reached similar conclusions [61,62,63]. Kim et al. [62] found that the satisfaction felt by students engaged in leisure activities leads to a higher level of well-being. Franzen et al. [63] highlight the impact of satisfaction on the psychological well-being of “students of health disciplines”. The satisfaction of students regarding the relationship they have with their professors and colleagues, but also the facilitating conditions in the university, directly influence their well-being, which is shown in Figure 2. Milmeister et al. [64] showed that both satisfaction in life and learning satisfaction influence the students’ well-being, which proves helpful for professors and academic managers who might adjust the strategies they use in the educational process.

The H5 hypothesis was supported. Thus, engaging in activities during their leisure time has a direct and positive impact on students’ well-being. Brajša-Žganec et al. [65] conducted a study in Croatia, proving the direct relationship between leisure activities and the respondents’ well-being. Another study by Shin and You [66] showed similar results with differences between genders (males preferring sports activities more than females). In our study, we included sports activities, but also volunteering, socializing, or relaxing activities. Other authors measured only the impact of these activities. Thus, Rodríguez-Bravo et al. [67] showed that sports activities positively impact the youngsters’ well-being. Pressman et al. [68] analyzed ten leisure activities and showed that they are related to a higher level of both psychological and physical well-being. Similar findings were reached by Han and Patterson [69].

The H6 hypothesis was supported, showing that self-exploration regarding their careers has a direct and positive impact on students’ well-being. Ran et al. [40] showed that both career exploration and self-reflection regarding their careers influence positively adolescents’ well-being. Zhou et al. [41] show that career exploration and career decision-making self-efficacy influence well-being. Still, the authors do not distinguish between career exploration oriented towards the exterior (environmental exploration) and the one oriented towards the self (self-exploration).

The H7 hypothesis was not supported. Thus, environmental exploration regarding their careers did not show to have a direct influence on students’ well-being. This hypothesis focused on the outside orientation of students in exploring their careers, referring to proactive behavior where they search for career opportunities and engage with professionals in their field. Other authors showed a correlation between career exploration in general [40,41] and well-being and also between proactive personalities and well-being [41]. In our study, more than 60% of the respondents are between 18 and 22 years old, which might explain why they are more focused on self-exploration regarding their career rather than environmental exploration. Ran et al. [40] showed that there is a direct relationship between career exploration and the students’ well-being, with the former being a predictor of the latter.

The H8 hypothesis was supported. The students’ well-being has a direct and positive impact on their academic engagement. Considering that well-being is influenced by so many factors (the support received from professors, family, and friends; leisure time activities; self-exploration regarding career; and the satisfaction felt), this finding is of utmost importance for both professors and academic managers who want to adjust the educational process in such a way that is conducive to a higher academic engagement and performance of students. Datu and King [42] showed that there is a reciprocal correlation between these two variables, meaning that not only does the well-being lead to academic engagement, but the latter also influences the different dimensions of students’ well-being. Eriksen and Bru [46] showed a positive correlation between emotional well-being and academic engagement, highlighting the role of positive emotions in the educational process, which might be considered by professors when they teach and evaluate students.

The present paper shows that the students’ well-being is influenced by many factors, such as the support received from their professors, their families, friends, and colleagues, but also by activities they enjoy doing in their free time (sports activities, relaxing, spending time in nature, going out with friends). The fact that well-being is responsible for the students’ academic engagement should convince professors and academic managers to consider the role played by well-being in the educational process and adjust the strategies used in teaching and managing the educational system. The academic results are better when students feel more satisfied and happier. The findings in this paper are helpful for educational management in shaping better strategies that focus not only on teaching or providing information and knowledge to students, but also on the way professors communicate with their students, listen to them, and encourage them in their efforts.

## 6. Conclusions

This research constitutes a robust exploration into the multifaceted factors influencing students’ well-being and academic outcomes, presenting nuanced findings that contribute significantly to the existing literature and theoretical frameworks. Despite not supporting the initial hypotheses suggesting a direct and negative impact of stressors in students’ lives and a direct and positive impact of environmental exploration on their well-being, this study unearthed crucial insights. Notably, the empirical evidence underscores the pivotal role of professors’ support in fostering students’ well-being, shedding light on the crucial relationship between academic mentorship and student mental health. Furthermore, the positive impact of social support from diverse sources stands out as a key determinant in shaping students’ overall well-being. The revelation that students’ perceived satisfaction in their lives serves as a direct and positive influence on well-being reinforces the interconnectedness of personal contentment and academic success. Additionally, this study brings to the forefront the significance of leisure activities and self-exploration in the context of career choices, both identified as direct and positive contributors to students’ well-being. Perhaps most notably, the reciprocal relationship established between students’ well-being and academic engagement advances our understanding, emphasizing that positive well-being not only correlates with but also positively influences academic commitment. This intricate interplay revealed in this study provides educators, policymakers, and researchers with actionable insights for the development of holistic student support systems and underscores the need for a comprehensive approach to student well-being and academic success.

The limits of our study are related to the fact that the questionnaire includes self-reporting opinions, which means that the answers reflect the students’ perception of their well-being, satisfaction, and academic engagement. Also, conducting a study in four countries is challenging, considering the differences that exist in their educational systems. Regarding future research directions, we appreciate that a study on the professors’ well-being will be helpful in better understanding the two main actors in the higher education system. Future research endeavors could delve deeper into the dynamics of stressors in students’ lives, aiming to identify specific stressor categories or contexts that exert varying effects on well-being. Exploring the intersectionality of stressors, such as academic pressures, personal challenges, and socio-economic factors, may provide a more nuanced understanding of their impact. Additionally, investigating the moderating factors that might mitigate or exacerbate the negative effects of stressors on well-being could offer valuable insights for intervention strategies. Further studies could also explore the temporal aspects of environmental exploration regarding careers, considering how students’ career aspirations evolve and how these changes influence their well-being. Research examining the specific mechanisms through which professors’ support and social support contribute to well-being would be beneficial, as well as exploring potential cultural or contextual variations in these relationships. Future investigations might also consider longitudinal designs to trace the long-term effects of leisure activities, self-exploration, and perceived satisfaction in students’ lives on their well-being and academic outcomes. Such endeavors would not only expand the knowledge base but also provide practical implications for educational institutions aiming to enhance students’ holistic development.

## Figures and Tables

**Figure 1 healthcare-12-01492-f001:**
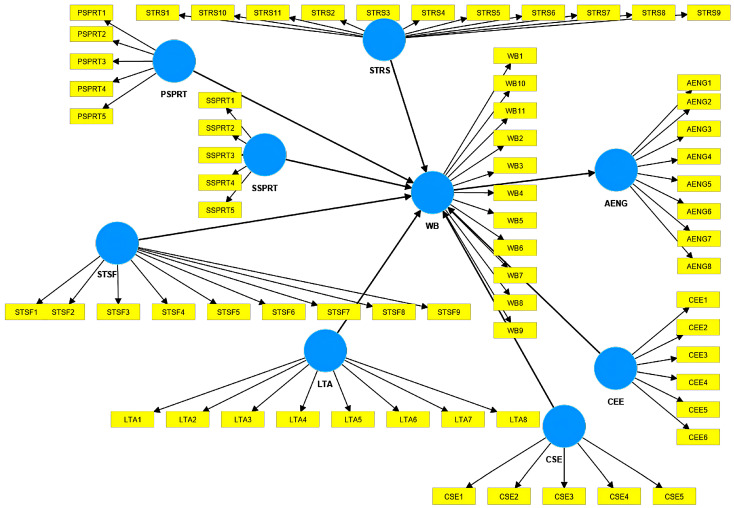
The research model generated in SmartPLS [47]. Note: stressors (STRS), well-being (WB), professors’ support (PSPRT), social support (SSPRT), satisfaction (STSF), academic engagement (AENG), leisure time activities (LTA), self-exploration regarding career (CSE), and environmental exploration regarding career (CEE).

**Figure 2 healthcare-12-01492-f002:**
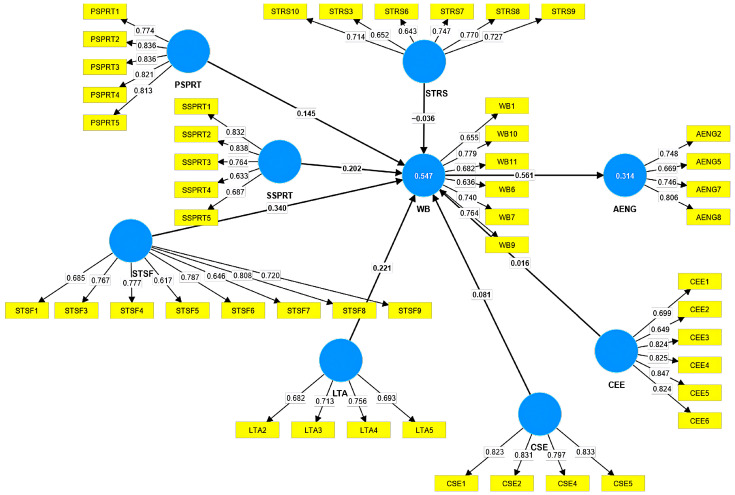
The research model with items exhibiting outer loadings higher than 0.6 [47]. Note: stressors (STRS), well-being (WB), professors’ support (PSPRT), social support (SSPRT), satisfaction (STSF), academic engagement (AENG), leisure time activities (LTA), self-exploration regarding career (CSE), and environmental exploration regarding career (CEE).

**Table 1 healthcare-12-01492-t001:** Characteristics of the sample.

Characteristics		Romania	Turkey	Slovakia	Bulgaria	All Countries
Sex	Male	44.9%	46.3%	29%	44.4%	38.7%
	Female	55.1%	53.7%	71%	55.6%	61.3%
Age	18–22 years old	46.3%	80.6%	62.4%	59.3%	60.4%
	23–27 years old	19.6%	10.3%	31.1%	11.7%	21.4%
	More than 27 years old	34.1%	9.1%	6.5%	29%	18.2%
Level of studies	Bachelor	79.4%	91.4%	67.5%	72.2%	75.6%
	Master	20.3%	7.4%	31.8%	24.1%	23.3%
	PhD	0.3%	1.2%	0.7%	3.7%	1.1%
Employment status	Yes	55.7%	19.4%	39%	63%	44.1%
	No	44.3%	80.6%	61%	37%	55.9%

**Table 2 healthcare-12-01492-t002:** The constructs and the items of the research model.

Constructs	Items	Codes
Stressors in students’ lives (STRS)—created by authors	I feel pressured to get high grades on tests and exams.	STRS1
	I feel pressured to engage in extracurricular activities.	STRS2
	I have financial difficulties.	STRS3
	I have family problems.	STRS4
	I have many commitments besides those related to the faculty.	STRS5
	Our professors give us too many assignments during the semester.	STRS6
	The study resources made available by the faculty are limited.	STRS7
	The quality of the educational services (courses/seminars/laboratories) is low.	STRS8
	The conditions provided by the university are under my expectations.	STRS9
	The quality of the administrative services offered by the university is low.	STRS10
	I feel harassed/discriminated against at the university.	STRS11
Students’ well-being (WB)—created by authors by using the three-dimensional scale mentioned by Supranowicz and Paz [13]	In general, my physical health is a good one.	WB1
	I am under treatment for a chronic condition/disease.	WB2
	My level of stress is high.	WB3
	I feel exhausted at the end of a day spent at the faculty.	WB4
	I have time also for activities outside the faculty.	WB5
	I do not have a problem focusing during courses/seminars/laboratories.	WB6
	I feel I can handle all the problems I encounter.	WB7
	I often have insomnia.	WB8
	I consider myself a happy person, in general.	WB9
	I have good relations with most of my professors.	WB10
	I have good relations with most of my colleagues.	WB11
Professors’ support (PSPRT)—created by authors	The professors trust my abilities.	PSPRT1
	My professors encourage me to express my opinions and engage in various activities.	PSPRT2
	My professors offer me constructive feedback when I make mistakes.	PSPRT3
	My professors are willing to help me if I ask for their help.	PSPRT4
	My professors express their appreciation towards me when I have great results.	PSPRT5
Social support from family, friends, and colleagues (SSPRT)—created by authors	I can count on my family’s support when I need it.	SSPRT1
	My family always encourages me to develop personally and professionally.	SSPRT2
	My family is interested in my student activity.	SSPRT3
	I can count on my colleagues.	SSPRT4
	I have friends I can count on when I need them.	SSPRT5
The students’ perceived satisfaction (STSF)—created by authors	In general, I am satisfied with my activity at the faculty.	STSF1
	I am satisfied with the relationship I have with my colleagues.	STSF2
	I am satisfied with the way my professors teach.	STSF3
	I am satisfied with the way my professors evaluate my activity.	STSF4
	I am satisfied with the relationship I have with the administrative office.	STSF5
	I am satisfied with the conditions at the university.	STSF6
	I am satisfied with the way my university ensures conditions for people with disabilities.	STSF7
	I feel proud to study at this faculty/university.	STSF8
	I like to spend time in the faculty/university even when I have a break between classes.	STSF9
Engaging in activities during their leisure time (LTA)—created by authors	Outside of faculty, I also engage in volunteering activities.	LTA1
	In my free time, I regularly practice sports.	LTA2
	In my free time, I take part in many relaxation activities.	LTA3
	In my free time, I like to spend time in nature.	LTA4
	In my free time, I like to hang out with friends.	LTA5
	In my free time, I like to socialize in a virtual environment.	LTA6
	In my free time, I like to spend time with my family.	LTA7
	In my free time, I dedicate myself to a hobby.	LTA8
Self-exploration regarding their careers (CSE) [37]	I reflected on how my past integrates with my future career.	CSE1
	I focused my thoughts on me as a person.	CSE2
	I contemplated my past.	CSE3
	I have been retrospective in thinking about my career.	CSE4
	I understood the new relevance of past behavior for my future career.	CSE5
Environmental exploration regarding their careers (CEE) [37]	I investigated career possibilities.	CEE1
	I went to various career orientation programs.	CEE2
	I obtained information on specific jobs or companies.	CEE3
	I initiated conversations with knowledgeable individuals in my career area.	CEE4
	I obtained information on the labor market and general job opportunities in my career area.	CEE5
	I sought information on specific areas of career interest.	CEE6
Academic engagement of students (AENG)—created by authors	I like to work more in a team than individually for the projects at the faculty.	AENG1
	Most of my grades are good/very good.	AENG2
	I engage in many extracurricular activities organized by my faculty/university.	AENG3
	I am actively involved in the student association of my faculty.	AENG4
	Besides the mandatory study materials, I also read additional ones.	AENG5
	I am present at most of my courses/seminars/laboratories.	AENG6
	I am very active during courses/seminars/laboratories.	AENG7
	I put passion and effort into the projects I have to do during the semester.	AENG8

**Table 3 healthcare-12-01492-t003:** The model’s reliability and collinearity.

Items	Outer Loadings	VIF
AENG1	0.381	1.080
AENG2	0.663	1.328
AENG3	0.624	2.162
AENG4	0.566	2.103
AENG5	0.629	1.386
AENG6	0.526	1.496
AENG7	0.743	1.906
AENG8	0.741	1.638
CEE1	0.672	1.554
CEE2	0.665	1.474
CEE3	0.832	2.255
CEE4	0.837	2.296
CEE5	0.850	2.655
CEE6	0.813	2.307
CSE1	0.825	1.885
CSE2	0.840	1.799
CSE3	0.560	1.675
CSE4	0.776	2.286
CSE5	0.830	1.911
LTA1	0.447	1.168
LTA2	0.635	1.377
LTA3	0.679	1.443
LTA4	0.706	1.483
LTA5	0.666	1.371
LTA6	0.167	1.067
LTA7	0.588	1.264
LTA8	0.578	1.281
PSPRT1	0.774	1.715
PSPRT2	0.838	2.202
PSPRT3	0.836	2.316
PSPRT4	0.817	2.066
PSPRT5	0.813	1.982
SSPRT1	0.824	3.456
SSPRT2	0.829	3.948
SSPRT3	0.757	2.128
SSPRT4	0.649	1.514
SSPRT5	0.694	1.574
STRS1	0.609	1.462
STRS10	0.667	1.589
STRS11	0.569	1.290
STRS2	0.509	1.384
STRS3	0.663	1.585
STRS4	0.585	1.424
STRS5	0.341	1.198
STRS6	0.622	1.459
STRS7	0.693	1.758
STRS8	0.705	2.020
STRS9	0.657	1.788
STSF1	0.696	1.590
STSF2	0.571	1.314
STSF3	0.757	2.273
STSF4	0.767	2.297
STSF5	0.604	1.511
STSF6	0.773	2.235
STSF7	0.635	1.498
STSF8	0.794	2.329
STSF9	0.714	1.820
WB1	0.657	1.602
WB10	0.695	1.830
WB11	0.614	1.611
WB2	−0.226	1.179
WB3	−0.523	1.878
WB4	−0.507	1.814
WB5	0.597	1.382
WB6	0.638	1.427
WB7	0.728	1.716
WB8	−0.407	1.299
WB9	0.742	1.767

**Table 4 healthcare-12-01492-t004:** Descriptive statistics for the model’s items.

Items	Mean	Standard Deviation
AENG2	3.562	1.051
AENG5	3.135	1.253
AENG7	3.310	1.134
AENG8	3.578	1.148
CEE1	3.924	1.103
CEE2	2.545	1.234
CEE3	3.078	1.325
CEE4	2.964	1.374
CEE5	3.193	1.322
CEE6	3.495	1.228
CSE1	3.641	1.165
CSE2	3.935	1.007
CSE4	3.668	1.144
CSE5	3.721	1.120
LTA2	2.973	1.347
LTA3	2.900	1.278
LTA4	3.641	1.213
LTA5	3.946	1.066
PSPRT1	3.530	1.008
PSPRT2	3.647	1.121
PSPRT3	3.713	1.075
PSPRT4	4.064	0.944
PSPRT5	3.710	1.089
SSPRT1	4.184	1.091
SSPRT2	4.185	1.091
SSPRT3	4.042	1.124
SSPRT4	3.473	1.215
SSPRT5	3.965	1.148
STRS10	2.684	1.299
STRS3	2.113	1.088
STRS6	3.489	1.313
STRS7	2.830	1.205
STRS8	2.789	1.268
STRS9	2.490	1.220
STSF1	3.487	1.046
STSF3	3.488	1.121
STSF4	3.529	1.079
STSF5	3.593	1.253
STSF6	3.330	1.205
STSF7	3.424	1.043
STSF8	3.507	1.166
STSF9	3.030	1.319
WB1	3.878	1.060
WB10	3.980	0.961
WB11	3.934	1.036
WB6	3.232	1.248
WB7	3.405	1.220
WB9	3.630	1.193

**Table 5 healthcare-12-01492-t005:** Reliability and validity of the model’s constructs.

Constructs	Cronbach’s Alpha	Composite Reliability (rho_a)	Composite Reliability (rho_c)	Average Variance Extracted (AVE)
AENG	0.732	0.744	0.832	0.553
CEE	0.870	0.878	0.903	0.611
CSE	0.842	0.862	0.892	0.674
LTA	0.676	0.677	0.804	0.506
PSPRT	0.874	0.875	0.909	0.666
SSPRT	0.807	0.809	0.868	0.570
STRS	0.804	0.803	0.859	0.505
STSF	0.873	0.879	0.900	0.531
WB	0.803	0.812	0.859	0.506

**Table 6 healthcare-12-01492-t006:** Fornell–Larcker criterion.

	AENG	CEE	CSE	LTA	PSPRT	SSPRT	STRS	STSF	WB
AENG	0.744								
CEE	0.385	0.782							
CSE	0.291	0.556	0.821						
LTA	0.327	0.317	0.266	0.712					
PSPRT	0.441	0.282	0.261	0.273	0.816				
SSPRT	0.345	0.174	0.139	0.395	0.463	0.755			
STRS	−0.195	−0.077	0.019	−0.139	−0.397	−0.219	0.710		
STSF	0.480	0.281	0.213	0.322	0.638	0.410	−0.579	0.729	
WB	0.561	0.306	0.286	0.481	0.555	0.518	−0.365	0.629	0.711

**Table 7 healthcare-12-01492-t007:** Bootstrapping test.

	T Statistics	*p* Value	Bias-Corrected Confidence Intervals	Hypotheses Validation
STRS → WB	1.299	0.194	(−0.088, 0.020)	H1 is not supported
PSPRT → WB	4.180	0.000	(0.077, 0.213)	H2 is supported
SSPRT → WB	6.925	0.000	(0.142, 0.256)	H3 is supported
STSF → WB	8.680	0.000	(0.265, 0419)	H4 is supported
LTA → WB	7.879	0.000	(0.166, 0.276)	H5 is supported
CSE → WB	2.872	0.004	(0.025, 0.135)	H6 is supported
CEE → WB	0.568	0.570	(−0.040, 0.071)	H7 is not supported
WB → AENG	23.472	0.000	(0.507, 0.603)	H8 is supported

## Data Availability

The data will be made available upon written request.

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
