# Peer review of "Students’ Well-Being and Academic Engagement: A Multivariate Analysis of the Influencing Factors"

_healthcare, 2024, doi:10.3390/healthcare12151492_

Round 1
Reviewer 1 Report
Comments and Suggestions for Authors
This paper discusses a very important topic, namely students’ well-being and academic engagement, and several factors. I have a couple of comments and questions that I would like the authors to address.
The study has been done in four countries: Romania, Turkey, Slovakia, and Bulgaria. This is not an obvious choice of countries, so I kindly ask the authors to explain why they chose exactly these countries. Also, discuss what makes these countries similar and what makes them different, and if this could have affected the results. It is mentioned that the interest was Eastern Europe, but there are other eastern European countries that were not included, and Turkey is not considered Europe, so the choice of countries should have a valid argument. Were the results different in different countries? If the authors hypothesized that there will not be any differences between the countries, what do they base that hypothesis on. In other words, what was the argument for combining the results from the different countries.
The authors put an emphasis on stress, but they don’t use it as their theoretical background. If the authors keep stress as their background, the theoretical models of stress should be explained.
What authors are describing in the paper is the positive psychology perspective, and especially positive youth development perspective, but that they explain the traditional perspective focused on negative experiences like stress. The authors are asked to position their paper in one of the perspectives and maintain that perspective consistently. I suggest describing positive youth development and building the paper on that perspective, since what they describe in the paper is actually PYD. If the authors refuse to include PYD, at least they could describe self-determination theory as a background for the influence of various types of support.
The authors differentiate life satisfaction from well-being while some models consider life satisfaction to be a component of well-being. The authors are asked to argue for their separation of life satisfaction from well-being, and to argue for the theoretical model they based their conceptualization of well-being.
The authors mention other studies that have investigated similar or the same relationships between the concepts included in this study. The authors are asked to describe the contribution of their study to the field. What new findings come from the study?
Pls is not a widely spread type of statistical analysis, so the authors are asked to argue for the need to use this analysis.
In the statistical analysis the authors assume that well-being affects academic engagement, while theoretically one could hypothesize the opposite direction. The authors are asked to argue why do they assume that well-being affects academic engagement and not the other way around. Or the bidirectional relation, like the Datu & King showed. Discussion of the bidirectional influence of all the concepts included in the study and well-being is advised.
The authors use “wellbeing” and “well-being” interchangeably. One of them should be used consistently, and grammatically correct is “well-being”.
The authors don’t describe how the data was collected. That is very important, and the authors are asked to provide more details on data collection procedure and informed consent.
And lastly, I am wondering if ethics approval in all countries was needed if the data were collected locally, and not just from one of the countries.
Comments on the Quality of English LanguagePhrasing reveals that the authors are not native English speakers and that they have use some tools to improve their writing.
Author Response
Dear Reviewer,
We are deeply grateful for the time and effort you have dedicated to analyzing our article. Your comments and recommendations have been invaluable, and we truly appreciate your astute observations that have helped us enhance the quality of our manuscript.
We sincerely appreciate all valuable observations and suggestions, which gave us a different perspective on our approach. In the following, we highlight your concerns and our efforts to address them. In response to your constructive criticisms, we have made significant revisions, which—we believe—improved the readability of our manuscript.
The revisions are highlighted in red in the manuscript:
Response to Reviewer 1 Comments
The study has been done in four countries: Romania, Turkey, Slovakia, and Bulgaria. This is not an obvious choice of countries, so I kindly ask the authors to explain why they chose exactly these countries. Also, discuss what makes these countries similar and what makes them different, and if this could have affected the results. It is mentioned that the interest was Eastern Europe, but there are other eastern European countries that were not included, and Turkey is not considered Europe, so the choice of countries should have a valid argument. Were the results different in different countries? If the authors hypothesized that there will not be any differences between the countries, what do they base that hypothesis on. In other words, what was the argument for combining the results from the different countries.
The decision to use these countries is based on the support we received within the Advtech_AirPollution project (Applying some advanced technologies in teaching and research about air pollution, 2021-1-RO01-KA220-HED-000030286) funded by the European Union within the framework of the Erasmus+ Program in which the universities from these countries were partners which made possible for us to investigate the students’ self-reported well-being.
Turkey has European and Asian territories. The universities from Turkey can be partners or coordinators in the projects funded by the European Commission. All countries involved in this project belong to South-East Europe. In the framework of an EU project, project partners have equal rights and will not make any difference between them.
- The authors put an emphasis on stress, but they don’t use it as their theoretical background. If the authors keep stress as their background, the theoretical models of stress should be explained.
What authors are describing in the paper is the positive psychology perspective, and especially positive youth development perspective, but that they explain the traditional perspective focused on negative experiences like stress. The authors are asked to position their paper in one of the perspectives and maintain that perspective consistently. I suggest describing positive youth development and building the paper on that perspective, since what they describe in the paper is actually PYD. If the authors refuse to include PYD, at least they could describe self-determination theory as a background for the influence of various types of support.
Thank you for your suggestion. We added a new reference in the literature review in the section for stressors that explains the correlation between PYD, stress, and well-being.
- The authors differentiate life satisfaction from well-being while some models consider life satisfaction to be a component of well-being. The authors are asked to argue for their separation of life satisfaction from well-being, and to argue for the theoretical model they based their conceptualization of well-being.
We used the three dimensions of well-being (physical, mental and social). Life satisfaction is a factor influencing well-being and its dimensions but they are not the same, as we show in the literature review, sections 2.1 and 2.5. We added in the text an explanation regarding the concept of well-being and its dimensions. For the three-dimension scale of well-being, we added the reference:
Supranowicz, P. and Paz, M., 2014. Holistic measurement of well-being: psychometric properties of the physical, mental and social well-being scale (PMSW-21) for adults. Roczniki Państwowego Zakładu Higieny, 65(3).
There are other studies too that we included in the paper measuring the impact of satisfaction on well-being, such as:
Urquijo, I.; Extremera, N.; Villa, A. Emotional intelligence, life satisfaction, and psychological well-being in graduates: The mediating effect of perceived stress. Applied research in quality of life 2016, 11; Ruiz‐
Aranda, D.; Extremera, N.; Pineda‐Galan, C. Emotional intelligence, life satisfaction and subjective happiness in female student health professionals: the mediating effect of perceived stress. Journal of psychiatric and Mental Health Nursing 2014, 21(2).
- The authors mention other studies that have investigated similar or the same relationships between the concepts included in this study. The authors are asked to describe the contribution of their study to the field. What new findings come from the study?
We expanded in a separate section as other reviewers also suggested the Theoretical model, its novelty, and usefulness deriving from our findings. This can be found in the newly added subsection 3.4. Theoretical model. The model is novel because it includes multiple factors at the same time that can impact well-being and indirectly academic engagement. Another element of novelty is related to the inclusion of self-exploration and environmental exploration of career choices in this complex model. It is helpful for academic managers, professors, and career counselors in universities.
- Pls is not a widely spread type of statistical analysis, so the authors are asked to argue for the need to use this analysis.
We added the explanation and cited a reference for choosing this method in our case. The added text is in the new subsection 3.3 Data Analysis.
- In the statistical analysis, the authors assume that well-being affects academic engagement, while theoretically one could hypothesize the opposite direction. The authors are asked to argue why do they assume that well-being affects academic engagement and not the other way around. Or the bidirectional relation, like the Datu & King showed. Discussion of the bidirectional influence of all the concepts included in the study and well-being is advised.
Our research focuses on Academic engagement as a dependent variable, our objective being to understand if and how (through the other variables affecting WB) well-being affects the students' engagement. When using PLS-SEM, you can test bidirectional causalities by building two or more models if you have a complex model like ours. Datu and King use a different method than us and they have a simpler model with only two variables that allowed them to have a bidirectional check.
So, a bidirectional causality is not possible for our model but we decided to test the correlation from AENG to WB (by creating a new model) as you suggested and the path coefficient is less than half (0.234) of the path coefficient from WB to AENG (0.561). We cannot add this information in the paper because it would imply to creation of multiple models which is not the aim of this paper.
- The authors use “wellbeing” and “well-being” interchangeably. One of them should be used consistently, and grammatically correct is “well-being”.
We changed accordingly. Thank you for noticing our mistake.
- The authors don’t describe how the data was collected. That is very important, and the authors are asked to provide more details on the data collection procedure and informed consent. And lastly, I am wondering if ethics approval in all countries was needed if the data were collected locally, and not just from one of the countries.
We added more details about the sampling procedure in subsection 2.2 and added that all respondents gave their consent before submitting their responses. There was no personal data collected. We decided to have the approval of the Ethics Committee for the leading partner university in the project and this research because the leading university created the survey that was applied in the other countries of the partnership. Another important detail, there are partnership agreements between the project leader and partners, signed by the Rectors of all four involved universities, that allow the researchers to exchange data related to the target group and research. We also added the acknowledgment to explain the context for this research and our choice for the four countries.

Reviewer 2 Report
Comments and Suggestions for Authors
In order to publish, I propose the following changes:
1) In lines 106-127 ("interventions to develop optimism"), 113 ("promoting teachers supportive behavior"), 127 ("critical"), the authors use quotation marks unnecessarily. They are not quotations from the text, but just expressions. Please correct.
2) On lines 125-126, there is a text quote in quotation marks with an author (Holliman et al.) which justifies the page indication. Please correct.
3) Figure 1 and Figure 2 can be combined into one to make the text shorter. The use of color is not justified. References to Figures 1 and 2 in the text should be adjusted to this change. Please correct.
4) The origin of the items used to measure the latent variables (Table 1) should be identified. If they are original, this should be mentioned. Please add this information.
5) Inserting several tables with secondary information makes the text too long and heavy. Table 1 can be sent as an supplement, or preferably integrated with Table 3. Please consider this possibility.
6) Table 3 can be reduced to indicating only the items selected to measure the latent variables and to build the model, for the sake of parsimony and convenience for the reader. If this suggestion is accepted, the origin of the items and the criteria used to select them for the study should be explained in the text. Please consider this possibility.
7) Between lines 251-255, many acronyms are used, which makes it difficult to understand the meaning of the text. It is suggested that the authors designate the latent variables without using acronyms. Please consider the suggestion.
8) In the final references, capitalize each of the words that designate the name of the journals (for example, references [2], [27], [28], [50], [60]). Please correct.
9) In the final references, the use of capital letters in the different words of the article titles should be eliminated (e.g. references [11], [12], [30], [31], [38], [42] , [51], [52]). Please correct.
10) Insert the pages in reference [11]. Please correct.
11) Complete the name of the journal in reference [53], the name of the journal is abbreviated. Please correct.
12) It would be didactic to include information on the guarantees given for anonymity and confidentiality about subjects and data, as well as on the procedure used to obtain informed consent in the different countries where the sample was collected.
Author Response
Dear Reviewer,
We are deeply grateful for the time and effort you have dedicated to analyzing our article. Your comments and recommendations have been invaluable, and we truly appreciate your astute observations that have helped us correct minor errors and enhance the quality of our manuscript.
We sincerely appreciate all valuable observations and suggestions, which gave us a different perspective on our approach. In the following, we highlight your concerns and our efforts to address them. In response to your constructive criticism, we have made significant revisions, which—we believe—improved the readability of our manuscript.
All revisions are highlighted in red in the manuscript:
Response to Reviewer 2 Comments
In order to publish, I propose the following changes:
- In lines 106-127 ("interventions to develop optimism"), 113 ("promoting teachers supportive behavior"), 127 ("critical"), the authors use quotation marks unnecessarily. They are not quotations from the text, but just expressions. Please correct.
Thank you for your observation. I used them exactly from those authors. Even if they seem like common expressions, we would prefer if you agree to keep them as such in order to avoid problems with those authors and plagiarism checks.
- On lines 125-126, there is a text quote in quotation marks with an author (Holliman et al.) which justifies the page indication. Please correct.
Thank you. We added the page for the text quote.
- Figure 1 and Figure 2 can be combined into one to make the text shorter. The use of color is not justified. References to Figures 1 and 2 in the text should be adjusted to this change. Please correct.
Thank you for your suggestion. As the Academic Editor suggested, we kept Figure 1 in the Methodology section where we were asked to present the research model. Thus, we cannot combine the figures. Figure 2 is part of the results section and is different than the initial research model in Figure 1. Both figures are generated by SmartPLS software cited in the text and we would prefer to not change the colors specific to the software used. We hope you understand and agree with our decision.
- The origin of the items used to measure the latent variables (Table 1) should be identified. If they are original, this should be mentioned. Please add this information.
We added the references in Table 1 and specified where the items are original. Thank you for your useful suggestion.
- Inserting several tables with secondary information makes the text too long and heavy. Table 1 can be sent as an supplement, or preferably integrated with Table 3. Please consider this possibility.
We understand your point of view. Still, we were advised by the Academic Editor to add several subsections in section 2 and detail Tables 1 and 2 in those subsections. Now, Table 1 became 2, and Table 2 became 1 to fit the new subsections we were asked to add. Table 1 is under subsection 3.1 Participants, and Table 2 is under subsection 3.2 Procedures and Measures.
- Table 3 can be reduced to indicating only the items selected to measure the latent variables and to build the model, for the sake of parsimony and convenience for the reader. If this suggestion is accepted, the origin of the items and the criteria used to select them for the study should be explained in the text. Please consider this possibility.
Thank you for your suggestion. We really appreciate that the Table might be too long, but because our model is so complex, we have no choice. Determining outer loadings and VIF values for all items in the model is a specific stage when using SmartPLS and we prefer to let all values in the table for transparency. It is important to show which items have adequate outer loadings and for the reader to understand the decision to give up some items, which we explained below the table.
We hope you understand our decision.
- Between lines 251-255, many acronyms are used, which makes it difficult to understand the meaning of the text. It is suggested that the authors designate the latent variables without using acronyms. Please consider the suggestion.
Thank you for your suggestions. When using PLS-SEM and SmartPLS, using acronyms is the norm and they are used as such in all studies using this method. They are described in Table 2 and the acronyms are used for simplicity. We hope you understand our perspective.
- In the final references, capitalize each of the words that designate the name of the journals (for example, references [2], [27], [28], [50], [60]). Please correct.
Thank you for your noticing our mistakes. We made the corrections. The references have now a different number because we added a few more references in the review process.
- In the final references, the use of capital letters in the different words of the article titles should be eliminated (e.g. references [11], [12], [30], [31], [38], [42] , [51], [52]). Please correct.
Thank you for noticing. We made the corrections.
- Insert the pages in reference [11]. Please correct.
Thank you for noticing this. We added the pages of the article.
- Complete the name of the journal in reference [53], the name of the journal is abbreviated. Please correct.
Now the reference is 56. We made the correction. Thank you for noticing.
12) It would be didactic to include information on the guarantees given for anonymity and confidentiality about subjects and data, as well as on the procedure used to obtain informed consent in the different countries where the sample was collected.
We included in the paper the number of the approval from the Ethics Committee. We also added a detailed explanation in section 3.2 (that we were asked by the Academic Editor to add) where we explain that respondents gave their informed consent and there was no personal data collected. The answers were anonymous so no other guarantees were needed.

Reviewer 3 Report
Comments and Suggestions for Authors
This is a study on students' well-being and academic engagement. This topic has certain significance, but in view of the limitations of the existing version, I think a lot of modifications are needed to ensure that it can be accepted.
The first is about the introduction. Although the authors have pointed out the limitations of relevant research, the reasons for the need for country-specific comparison and the problems of this study are not clear.
Secondly, it is about literature review. The authors start from the study of several variables, but lack effective theoretical basis and the relationship between variables. Therefore, the research hypothesis put forward is actually unfounded.
Thirdly, about pictures. The picture presentation in the article can be said to be very bad. If it can't be solved, the manuscript will have to be rejected, so the authors are advised to consider it carefully.
Fourthly, about Table 1. I suggest that the authors put the original title of the scale in the appendix, not in the text. Not only does it affect reading, but it is also because your typesetting is very confusing.
Fifthly, about the results. The presentation of the results is also very poor. The cross-sectional data used in this study have no basic reliability and validity and common method deviation test, and the related path coefficient has not been reported. It is suggested that all proofreading should be done according to the format of published research.
Finally, there are some mistakes in the grammar of this study, so please proofread them in detail.
Author Response
Dear Reviewer,
We are deeply grateful for the time and effort you have dedicated to analyzing our article. Your comments and recommendations have been invaluable, and we truly appreciate your astute observations that have helped us enhance the quality of our manuscript.
We sincerely appreciate all valuable observations and suggestions, which gave us a different perspective on our approach. In the following, we highlight your concerns and our efforts to address them. In response to your constructive criticisms, we have made significant revisions, which—we believe—improved the readability of our manuscript.
All revisions are highlighted in red in the manuscript.
Response to Reviewer 3 Comments
- The first is about the introduction. Although the authors have pointed out the limitations of relevant research, the reasons for the need for country-specific comparison and the problems of this study are not clear.
Thank you for your suggestions to improve our paper. As the other reviewers and the Academic Editor also suggested, we were asked to add more details on the participants in the newly added subsection 3.1 Participants under the Materials and Methods section. We understand that surveying students in four countries is challenging and we had the opportunity to investigate this important topic as part of the collaboration between universities in the four countries.
The decision to use these countries is based on the support we received within the Advtech_AirPollution project (Applying some advanced technologies in teaching and research about air pollution, 2021-1-RO01-KA220-HED-000030286) funded by the European Union within the framework of the Erasmus+ Program in which the universities from these countries were partners.
We also added a paragraph in subsection 3.2 to better highlight the novelty and usefulness of our research. We also acknowledge the limitations of the study in Conclusions.
- Secondly, it is about literature review. The authors start from the study of several variables, but lack effective theoretical basis and the relationship between variables. Therefore, the research hypothesis put forward is actually unfounded.
Thank you for your recommendation to improve our paper. We added a few more references to explain the variables better. The hypotheses were moved to the Literature review section.
The relationships between variables are discussed in the Discussion section where we introduced more details on the findings of other authors. The hypotheses are discussed in detail in section 5.
- Thirdly, about pictures. The picture presentation in the article can be said to be very bad. If it can't be solved, the manuscript will have to be rejected, so the authors are advised to consider it carefully.
It is possible that in the Word or pdf document you had access to, the quality is not so good. We will send the images at their higher resolution to the Academic Editor. Thank you for your suggestion.
- Fourthly, about Table 1. I suggest that the authors put the original title of the scale in the appendix, not in the text. Not only does it affect reading, but it is also because your typesetting is very confusing.
Thank you for your suggestion. We were asked by the Academic Editor to add subsections regarding the Participants, Procedure and the measures, Data Analysis and Theoretical Model under section 3 and keep Table 1 (former Table 2) and Table 2 (former Table 1) there in the relevant subsections. We hope you understand.
The formatting of the table is specific to the journal and the abbreviations used are codes specific to the items and variables in the model. They are in line with other studies using PLS-SEM.
- Fifthly, about the results. The presentation of the results is also very poor. The cross-sectional data used in this study have no basic reliability and validity and common method deviation test, and the related path coefficient has not been reported.
The results follow all steps when using research methodology PLS-SEM and SmartPLS. We added more explanations in the Method section about PLS-SEM and also in the section 3.4 Theoretical Model. The path coefficients are presented in Figure 2 and all statistical tests are correct. We used Smart-PLS version 4.1.0.0.
The model’s reliability and collinearity are presented in Table 3. The reliability and validity of the constructs in the model are determined in Table 4. The Cronbach’s alpha is higher than 0.7 for 8 of the 9 constructs, but the value for LTA is considered acceptable [51,52], being higher than 0.6 and close to 0.7. The same is noticed for the composite reliability. The Standardized Root Mean Square Residual (SRMR) is 0.06 which is lower than the threshold of 0.08 for the model to be considered a good fit.
All tables are explained in the text. The presentation of results is in line with other studies using the method of PLS-SEM.
We also improved the Discussion section.
- Finally, there are some mistakes in the grammar of this study, so please proofread them in detail.
Thank you for helping us improve our manuscript. We double-checked the grammar.
We appreciate your time and effort in reviewing our paper.

Reviewer 4 Report
Comments and Suggestions for Authors
The topic of this study is relatively outdated, and there have been too many studies that have focused on this issue. The literature review still needs significant improvement. In addition, the format of the paper is not standardized and needs improvement. The research conclusions and discussions are also not comprehensive enough.
The literature review is somewhat inappropriate, resembling a mere listing of references rather than being organized around the research questions. The author should summarize several theoretical perspectives based on previous studies, then discuss each of these perspectives separately. Following this, they should propose an analytical framework for the current study and subsequently derive research hypotheses. At present, the study lacks both an analytical framework and a well-reasoned or sufficiently supported logical derivation for the hypotheses. The literature review section needs to be restructured.
Placing the research hypotheses in the methodology section is completely incorrect. The methodology section should include the specific methods used in the study, such as data collection methods, data analysis methods, the specific statistical analysis models employed, and the software used.
The study lacks clarification on the specific variables, including what the dependent variable is, which independent variables are included, and what the control variables are. Additionally, there is no explanation of how these variables are measured, nor is there a description of the basic data and variables. There is a lack of necessary clarification on what is well being and academic engagement. In addition, the relationship between the two also needs to be explained clearly.
In the results, the path analysis result is presented without any descriptive analysis, usually, descriptive analysis is needed before further stastical analysis.
The discussion in this study is not sufficiently in-depth. The discussion of the research results should not only revolve around the outcomes of each specific hypothesis. In addition to a detailed comparison with previous research findings, highlighting agreements, disagreements, and possible reasons, there should also be a comprehensive discussion to emphasize the overall contributions of the study.
Overall, this is an unregulated and immatured research, and there is still room that need improvement before being published .
Author Response
Dear Reviewer,
We are deeply grateful for the time and effort you have dedicated to analyzing our article. Your comments and recommendations have been invaluable, and we truly appreciate your astute observations that have helped us enhance the quality of our manuscript.
We sincerely appreciate all valuable observations and suggestions, which gave us a different perspective on our approach. In the following, we highlight your concerns and our efforts to address them. In response to your constructive criticism, we have made significant revisions, which—we believe—improved the readability of our manuscript.
All revisions are highlighted in red in the manuscript.
Response to Reviewer 4 Comments
- The literature review needs significant improvement. In addition, the format of the paper is not standardized and needs improvement. The research conclusions and discussions are also not comprehensive enough.
Thank you for your valuable suggestions. We added more references in the Literature review section for well-being and its three dimensions and for the stressors. We moved the Hypotheses after presenting the main variables that represent the foundation of our research model.
We added more subsections in the Methodology to be in line with the format specific to the journal. Subsections 3.1 to 3.4 were added.
The literature review presents the variables of the research model we used. We also improved the Discussion section to present in-depth the relationship between the variables in the literature review and emphasize the findings reached by other authors.
We eliminated some redundant phrases from the Conclusion to make the text clearer.
- The literature review is somewhat inappropriate, resembling a mere listing of references rather than being organized around the research questions. The author should summarize several theoretical perspectives based on previous studies, and then discuss each of these perspectives separately. Following this, they should propose an analytical framework for the current study and subsequently derive research hypotheses. At present, the study lacks both an analytical framework and a well-reasoned or sufficiently supported logical derivation for the hypotheses. The literature review section needs to be restructured.
We appreciate your perspective on this matter. For this study, we proposed a research model which is described in Section 3. When applying PLS-SEM as a research methodology, the literature review is organized around the variables in the model. The presentation is in line with other papers using this method.
We improved the section by adding a few more references in the Literature review section and also in Methodology to explain our choice for PLS-SEM.
We moved Hypotheses after the variables presented in the literature review in subsection 2.9.
We presented the relationships between variables and discussed the hypotheses in section Discussions in more detail, elaborating on the findings of other authors.
- Placing the research hypotheses in the methodology section is completely incorrect. The methodology section should include the specific methods used in the study, such as data collection methods, data analysis methods, the specific statistical analysis models employed, and the software used.
Thank you for your astute observation. Now, the Methodology section is completely restructured. We added 4 subsections and more details to explain the data collection, analysis, and the research model used.
- The study lacks clarification on the specific variables, including what the dependent variable is, which independent variables are included, and what the control variables are. Additionally, there is no explanation of how these variables are measured, nor is there a description of the basic data and variables. There is a lack of necessary clarification on what is well-being and academic engagement. In addition, the relationship between the two also needs to be explained clearly.
We added subsections 3.3 and 3.4 to better explain PLS-SEM and the theoretical model we used. Figures 1 and 2 show the dependent (WB and AENG) variables and the path coefficients for the relationships with the independent variables. We added a detailed explanation in the Methodology section. Subsection 3.2 Procedure and Measures explains the scale we used for the items illustrated in Table 2. Table 2 (former Table 1) is now improved and we added references for the constructs in the model where it was the case.
We added a reference in the literature review to clarify the well-being dimensions (physical, mental, and social), and the concept of academic engagement is presented in the literature review. The relationship between the two is presented in subsection 2.8 by references 42 to 46. Also, in the Discussion section, the last paragraph presents the findings of other authors regarding the impact of well-being on academic engagement.
- In the results, the path analysis result is presented without any descriptive analysis, usually, descriptive analysis is needed before further statistical analysis.
Thank you for your valuable suggestion. We added a new table with the descriptive statistics of the items in the model that have outer loadings above 0.6 because these were the ones kept in the model and we added the explanation for it.
- The discussion in this study is not sufficiently in-depth. The discussion of the research results should not only revolve around the outcomes of each specific hypothesis. In addition to a detailed comparison with previous research findings, highlighting agreements, disagreements, and possible reasons, there should also be a comprehensive discussion to emphasize the overall contributions of the study.
We added more explanations in the Discussion section to present the findings of other authors. Also, we moved the presentation of the overall contribution of our research from Conclusions to the end of Discussion, lines 412-422 to summarize the previous discussion of hypotheses.

Round 2
Reviewer 3 Report
Comments and Suggestions for Authors
I think picture is very bad, please refer to the relevant articles to modify it (See below). At the same time, it should be noted that you should add notes under the picture, and what the abbreviations in the picture represent.
eg
Seo, J.; Ko, H. Effects of Self-Leadership on Nursing Professionalism among Nursing Students: The Mediating Effects of Positive Psychological Capital and Consciousness of Calling. Healthcare 2024, 12, 1200. https://doi.org/10.3390/healthcare12121200
Author Response
Response to Reviewer 3 Comments
I think picture is very bad, please refer to the relevant articles to modify it. At the same time, it should be noted that you should add notes under the picture, and what the abbreviations in the picture represent.
Reply: Thank you for your recommendation.
- We inserted in the attached Word for authors’ reply the pictures at a higher quality (also in the manuscript) and we will send them to the editors in a separate archive.
Unfortunately, the platform does not allow us to send you JPEG or PNG files, but we will ask editors to send the files to you. Word and PDF comprise the quality of the images and we apologize for this inconvenience.
- The images are created in the specific software we used in methodology - SmartPLS. They are generated with this design by the software that calculates the path coefficients. This is why they are different from the example you kindly offered to us.
- We added a reference under the pictures for the software as can be seen in the manuscript.
The reference is number 47 - Ringle, C. M.; Wende, S.; Becker, J.-M. SmartPLS 4. Oststeinbek: SmartPLS GmbH, 2022. Available online at http://www.smartpls.com
- We also added a note with the abbreviations, as you suggested, for better clarity. The changes can be seen in the manuscript. We also, add the note here.
Note: Stressors (STRS), Well-being (WB), Professors’ support (PSPRT), Social support (SSPRT), Satisfaction (STSF), Academic engagement (AENG), Leisure time activities (LTA), Self-exploration regarding career (CSE), and Environment exploration regarding career (CEE).
Thank you for your understanding and patience.
We are deeply grateful for the time and effort you have dedicated to analyzing our article. Your comments and recommendations have been invaluable, and we truly appreciate your astute observations that have helped us enhance the quality of our manuscript.
